# Endosymbiotic Bacterial Diversity of Corn Leaf Aphid, *Rhopalosiphum maidis* Fitch (Hemiptera: Aphididae) Associated with Maize Management Systems

**DOI:** 10.3390/microorganisms10050939

**Published:** 2022-04-30

**Authors:** Artúr Botond Csorba, Ciprian George Fora, János Bálint, Tamás Felföldi, Attila Szabó, István Máthé, Hugh D. Loxdale, Endre Kentelky, Imre-István Nyárádi, Adalbert Balog

**Affiliations:** 1Department of Horticulture, Faculty of Technical and Human Sciences, Sapientia Hungarian University of Transylvania, Aleea Sighișoarei 2, 540485 Târgu Mureș, Romania; csorba.artur@ms.sapientia.ro (A.B.C.); balintjanos@ms.sapientia.ro (J.B.); kentelky@ms.sapientia.ro (E.K.); 2Faculty of Horticulture and Forestry, Banat’s University of Agricultural Sciences and Veterinary Medicine King Michael I of Romania from Timișoara, Calea Aradului 119, 300645 Timișoara, Romania; 3Department of Microbiology, Eötvös Loránd University, Pázmány Péter Sétány 1/c, 1117 Budapest, Hungary; tamas.felfoldi@gmail.com; 4Department of Aquatic Sciences and Assessment, Swedish University of Agricultural Sciences, Lennart 756-51 Hjelms Väg 9, 750 07 Uppsala, Sweden; attila.szabo.ttk@gmail.com; 5Department of Bioengineering, Sapientia Hungarian University of Transylvania, Piaţa Libertăţii 1, 530104 Miercurea Ciuc, Romania; matheistvan@uni.sapientia.ro; 6School of Biosciences, Cardiff University, The Sir Martin Evans Building, Museum Avenue, Cardiff, Wales CF10 3AX, UK; loxdale@web.de

**Keywords:** monoculture, large-scale crops, total bacterial diversity, primary symbionts, secondary symbionts, amplicon sequencing, 16S rRNA gene

## Abstract

In this study, different maize fields cultivated under different management systems were sampled to test corn leaf aphid, *Rhopalosiphum maidis*, populations in terms of total and endosymbiotic bacterial diversity. Corn leaf aphid natural populations were collected from traditionally managed maize fields grown under high agricultural and natural landscape diversity as well as conventionally treated high-input agricultural fields grown in monoculture and with fertilizers use, hence with low natural landscape diversity. Total bacterial community assessment by DNA sequencing was performed using the Illumina MiSeq platform. In total, 365 bacterial genera were identified and 6 endosymbiont taxa. A high abundance of the primary endosymbiont *Buchnera* and secondary symbionts *Serratia* and *Wolbachia* were detected in all maize crops. Their frequency was found to be correlated with the maize management system used, probably with fertilizer input. Three other facultative endosymbionts (“*Candidatus Hamiltonella*”, an uncultured *Rickettsiales* genus, and *Spiroplasma*) were also recorded at different frequencies under the two management regimes. Principal components analyses revealed that the relative contribution of the obligate and dominant symbiont *Buchnera* to the aphid endosymbiotic bacterial community was 72%, whereas for the managed system this was only 16.3%. When facultative symbionts alone were considered, the effect of management system revealed a DNA diversity of 23.3%.

## 1. Introduction

Maize (*Zea mays* L.) is the largest crop in Romania. Some 2 million ha are cultivated every year (average 18.6 million tons/year), making Romania the ninth largest producer in the world. One increasing problem in maize cultivation is the occurrence of the major pest of cereals, the corn leaf aphid, *Rhopalosiphum maidis* Fitch (Hemiptera: Aphididae) over the last 5 years, especially in Germany, Poland, Hungary, and from 2019 onwards, in Romania (Banat region), not only as a result of direct physical damage (i.e., sap-feeding), but also due to the transmission of plant pathogenic viruses [1,2,3].

Aphids are closely associated with microorganisms, the obligate mutualist endobacterium (usually referred to as a primary symbiont), *Buchnera aphidicola*, which is maternally inherited [4] as well as other maternally transmitted intracellular bacteria, such as *Rickettsia* spp. (α-Proteobacteria), *Spiroplasma* spp. (Mollicutes), and various γ-proteobacterial microbes (including *Hamiltonella defensa, Regiella insecticola, Serratia symbiotica*, and *Arsenophonus* spp.) [5]. These aphid secondary symbionts are often shared between divergent lineages and seem to undergo both vertical and horizontal transfer among matrilines [4]. Several previous publications suggest that these symbiotic bacterial communities are involved in the expression of different traits related to aphid biology, including resistance to parasitoid wasps [6], tolerance to heat stress [7], and changes in the host plant range [8]. Overall, the functional role of bacterial endosymbionts have been defined in relation to two main traits: (1) those that confer advantages to the host aphid under specific ecological conditions, often defined as a protective role against abiotic conditions (high temperatures) or natural enemies and (2) the particular role of the endosymbiont in relation to the aphid host’s metabolism, in terms of the specific nutrients that the endosymbionts synthesize and release and that are required by the host [9].

To date, few studies have been performed that throw light on how aphids can adapt to different environments via their symbionts. Previous studies have revealed through heat exposure experiments of *Buchnera* endosymbionts their role in the aphid host’s fitness parameters, including host survival, maturation time, and fecundity, and interestingly showed that different aphid species are differentially adapted to such treatment. Thus, for example, in the Cotton-melon aphid, *Aphis gossypii* Glover, heat exposure of the aphid enhanced its fecundity yet seemingly had no effect on the *Buchnera* titre as such. This contrasts with the situation found in the black bean aphid, *Aphis fabae* Scopoli, whereupon the aphid suffered high mortality following heat treatment due to suppression of the obligate *Buchnera* population, consequently leading to direct effects on the host’s development as well as reducing its fecundity. Lastly, in the blue alfalfa aphid, *Acyrthosiphon kondoi* Shinji, and the pea aphid, *A. pisum* Harris, heat exposure caused rapid declines in *Buchnera* symbiont abundances and reduced survivorship, development rate, and fecundity. Altogether, these experiments show conclusively that aphid fecundity following heat exposure is significantly influenced by its effect on *Buchnera*. It appears that aphid fecundity decreases due to a mutational change which suppresses the transcriptional response of a gene encoding a small heat shock protein. The absence of this heat shock gene conferred by *Buchnera* seems to explain the heat sensitivity of *A. fabae*. Altogether, these results demonstrate that changes in *Buchnera* heat sensitivity contribute to aphids’ heat tolerance [10].

In the case of the corn leaf aphid, however, the role of symbionts in the aphid’s adaptations to various environmental conditions (e.g., low vs. high complexity) and management systems (e.g., low vs. high input of chemicals) remains unclear. The aphid’s major host, plant maize, is cultivated under different management systems (high or low input of fertilizers and insecticides) and under different climatic conditions in Europe [1,11], which makes it a very good experimental system in which to investigate the role of symbionts in terms of the host’s adaptation under different field conditions, and hopefully elucidate the main ecological-environmental factors involved.

A previous study involving microsatellite analyses on the genetic diversity of *Aphis gossypii* collected over a large geographical scale showed that this aphid species harbours several host-associated microbial communities with wide distribution [12]. Other studies have also reported that the aphid genus *Cinara* represents a ‘global diversification scenario’, whereby speciation processes are strongly constrained by host genus associations [13]. The same research also revealed a pattern of frequent niche shifts in terms of host plant use and feeding habits. *Cinara* species as a group show frequent host specialization events and multiple transitions from branch-feeding to shoot-feeding. Similarly, many host-parasitoid interactions (especially those involving hymenopterous (wasp) parasitoids and fungi) are influential, but with unknown consequences, in the biological control of aphid pests [14]. This multitude of interacting factors (crop management, climate and land-use change, as well as natural enemies) that impinge upon aphids seemingly drives them towards rapid adaptation [15]. In Europe, damage by corn leaf aphids to maize occurs from June until the end of September [16], whilst the most favorable temperature for aphid development is between 28 and 30 °C with a relative humidity of 60–70%. The present paper represents the full results obtained by us following detailed investigation of this particular tritrophic system—plant–aphid–symbiont.

Objectives: In this light, the present study concerns our efforts to answer the central question, important both from a fundamental as well as potentially an applied point of view (i.e., aphid pest control): Does host plant (here maize) management, i.e., conventionally managed large and medium scale fields, with fertilizer and pesticide input vs. small scale conventionally managed low input field farming and local gardens, influence corn leaf aphid endosymbiotic bacterial community composition and diversity? If so, what does this reveal about the potential relevance of these symbionts in the life history and life cycle of this aphid species, from which we may perhaps be able to gain further insights on how these microorganisms can be manipulated to effect better pest control?

## 2. Material and Methods

### 2.1. Study Areas

From each management system, two maize fields were sampled and inside each one, two sample sites were defined. The two large-scale monoculture fields were situated in western Romania (Banat region, Timis County), located about 15 km apart. Medium-scale fields were located in central Romania (Sfantu Gheorghe city area) and were both represented by semi-intensive maize monocultures, located about 10 km apart. Small-scale maize fields were located in central Transylvania (Targu-Secuiesc city area), having a distance of about 8 km apart. Local gardens were located in the Sighisoara area (Saxon region) of central Transylvania, Romania, about 5 km from each other. This region comprises mostly valleys between mountains with no pesticide and fertilizer usage during maize production, characterized by a landscape mosaic of different land-cover types (approximately 28% forest, 24% pasture, and 37% arable land) (Table 1).

### 2.2. Field Collection of Corn Leaf Aphids

Aphid samples were collected from the four maize fields, both in two replicates during the same time period (June and July, 2020) when all plots were in flowering stages R2 to R3. Inside all maize fields, two smaller (sites) were sampled, except local gardens, from which only two separate fields were sampled. Aphids were collected from maize plants inside the field; this was done to minimize the possible effects of field margins. Similar sampling methods were used in all sites; asexual lineages of wingless (apterous) individuals were collected as follows: ten maize plants per semi-field were randomly selected at each sampling date, with first instar aphid nymphs (five from each colony/plant) collected and stored in 0.5 mL Eppendorf tubes containing 99% ethanol prior to DNA analysis.

### 2.3. Aphid-Associated Bacterial Community Analysis

The bacterial communities associated with aphids were studied with Illumina amplicon sequencing of the V3–V4 region of the 16S rRNA gene using OTUs (operational taxonomic units) defined at 97% nucleotide sequence similarity level. For DNA extraction and subsequent PCR amplification, aphids (five asexual individuals from each colony sampled/one semi field/field) were first washed twice with 70% ethanol to remove surface-attached microbes. Analysis was based on amplicon sequencing of the 16S rRNA gene, as in our previous works [17,18]. Briefly, total genomic DNA was extracted using the DNeasy PowerSoil Kit (Qiagen), whereafter a part of the 16S rRNA gene was amplified using primers with the bacteria-specific sequences Bakt_341F (5′-CCT ACG GGN GGC WGC AG-3′ [19]) and Bakt_805NR (5′-GAC TAC NVG GGT ATC TAA TCC-3′ [20]). DNA sequencing was performed on an Illumina MiSeq platform using MiSeq standard v2 chemistry as a service provided by the Genomics Core Facility RTSF, Michigan State University, USA.

### 2.4. Data Analyses

Methodological details and the applied bioinformatics and statistical analyses were as described in [17], except that the resulting sequence reads were processed using the mothur v1.41 software [21]. This was based on the MiSeq standard operating procedure (downloaded on 3 April 2020), with the removal of chimeric sequences performed using VSEARCH [22] and a taxonomic assignment based on the ARBSILVA SSU reference database [23]. Raw sequence data were submitted to NCBI under BioProject ID PRJNA647165.

For statistical analysis of amplicon sequencing data, the subsampling of reads was performed to the read number of the smallest dataset (*n* = 56,288). Microbial diversity indices and species richness values (using the Chao1 and the ACE richness metrics) were calculated using mothur v1.41.

The variation in total bacterial diversity indices (Simpson, Shannon, Evenness) were computed under different management systems using PAST v4.02 [24].

Relative abundance patterns of the endosymbiotic taxa present on all maize fields were performed at the level of taxonomic paths (resolved up to the genus level). Community patterns were compared by cluster analysis in PAST v4.02 using UPGMA as clustering algorithm and calculation of Bray–Curtis similarity indices. Endosymbiotic taxa were sorted by their abundance (total genomic DNA) in relation to the differences noted between managements systems. Data are shown for taxa with >0.1% contribution.

Principal components analyses (PCAs) were used to identify the proportion of variation in each PCA axis (bacterial DNA diversity and management). The average count of each bacterial DNA reads numbers detected and log_10_ transformed from each sample grouping were used as *response variables* designated as component 1 (PCA axis1) and *management systems* as component 2 (PCA axis 2) and explained by the most dominant bacteria related to each management regime (i.e., dominant bacteria DNA reads numbers, separated from *Buchera* dominance).

## 3. Results

Altogether, 365 genus-level bacterial taxa were identified related to corn leaf aphids under the four management systems (Appendix A online materials). The total bacterial diversity found by using linear regression analyses related to corn leaf aphid differed even within management systems; the highest taxonomic numbers were detected at large-scale farming field A12 with 166 species, while the lowest was also observed at large-scale farming site A11 with 28 species. An average of 50 to 53 species of total bacteria per management systems were detected during the study. Diversity indices also revealed that several bacterial species were found at low frequency, and only a few species of the total bacterial community were seen to be dominant (Table 2).

Considering only the endosymbiotic bacterial taxa, these varied between management systems, with six genera from six phyla being detected. The obligate symbiont *Buchnera* was present at all sites, but with different frequencies according to the total genomic DNA analyzed. Aphids collected from large-scale crops site A12 all had four species of bacterial endosymbionts, while aphids from the other locations often had fewer. Only the obligate symbiont *Buchnera* was found in all samples. At a phylum level, *Proteobacteria* were dominant in all aphid colonies (Figure 1A). Among symbionts, at genus level after the obligate symbiont *Buchnera*, the facultative symbiont *Serratia* was also frequent, followed by *Wolbachia*. *Serratia* was present in aphids collected from all large scale-crops, but it was also present in all management systems (even if missed in some plots) with lower frequency. *Wolbachia* were not detected in medium-scale crops and in gardens (Figure 1B). *Candidatus Hamiltonella* was present only in one large-scale crop. *Rickettsiales*, other than genus *Spiroplasma*, were detected at very low frequency in aphids in only one medium-scale maize field. *Spiroplasma* was present in aphids colonies collected from large-scale fields and from gardens (Figure 1B).

The relative abundance patterns of endosymbiotic taxa, as revealed at the level of taxonomic paths (resolved up to the genus level) and present on all maize fields, was seen to clearly relate to the management system involved. This, in turn, showed the dominance of *Buchnera* and *Serratia*, followed by *Wolbachia.* Using UPGMA as clustering algorithm and Bray–Curtis similarity, some effects of the management systems used can be detected; groups from the same management systems show similarity, but also some differences (Figure 2).

PCAs revealed the proportion of variation per each PCA axis (bacterial DNA diversity as axis 1, management systems as axis 2), which were explained by the most frequent obligate and facultative bacterial distribution. When the obligate and most dominant symbionts *Buchnera* are considered, the total DNA diversity represented 72% of the total, whereas management systems only represented 16.3% (Figure 3A). When only facultative symbionts were considered, the effect of management systems define distributions at a level of around 23.3%, while total DNA diversity define distributions at 43.8% (Figure 3B).

## 4. Discussion

In the present study, 365 bacterial genera were identified, of which six endosymbionts were detected and identified as being associated with corn leaf aphid populations infesting maize crops grown under different management systems. Other than our study, only a few previous published studies have presented the detailed species composition of endosymbiotic bacterial communities associated with particular aphid species [25].

Primary and secondary symbionts (*Buchnera* and uncultured members of the family Enterobacteriaceae) dominated these communities in all maize fields and management systems. *Buchnera* is generally required for the survival of aphids and provides essential amino acids that are rare in their phloem sap diet [26] but also have a significant role in aphids’ heat tolerance [10]. The facultative endosymbiotic *Wolbachia* dominated only one large-scale crop and was also present in small-scale fields. Furthermore, *Serratia* was relatively more frequent in large-scale crops than in other crops. *Candidatus Hamiltonella* was present only in one large-scale site. To our knowledge, this is the first description of the diversity of endosymbiotic bacteria inhabiting corn leaf aphids infesting maize crops under different management systems. By using a high resolution molecular (DNA) approach, it is clear that whilst the aphids are not ‘free agents’ in terms of their biology and lifestyle, including host plant adaptation [11,27,28], nor indeed are the bacterial endosymbionts inhabiting these insects free of constraints (i.e., management systems including plus or minus synthetic fertilizer and insecticide application). Hence, these factors may well have—and indeed probably do have—a significant influence on endosymbiotic bacterial diversity.

Other factors related to the adaptation of corn leaf aphid via endosymbionts however need to be considered and further studied. It has been reported that almost all aphids are hosting the obligated and anciently acquired endosymbiont *Buchnera*, although some recent studies reported that *Geopemphigus* aphids have lost the obligated symbionts *Buchnera*, which was relocated with a maternally-transmitted symbiont from the bacterial phylum *Bacteroidetes* [29]. The novel symbiont found in *Geopemphigus* aphids that functionally and physically replaced *Buchnera* was named “Ca. Skilesia alterna” [30]. If symbiotic bacterial species are no longer providing functions to their hosts, the host may acquire additional symbiont species or even replace the existing symbiont with a new one/s [29]. This was previously demonstrated in aphids within the tribe *Cerataphidini* (subfamily *Hormaphidinae*), a group not closely related to *Geopemphigus* aphids. These species of *Cerataphis* harboured an extracellular fungal symbiont that lives in the body cavity [31,32]. Both *Cerataphidini* and *Geopemphigus* are adapted to warm climates and because *Buchnera* is sensitive to high temperatures, it is possible that aphids living in these hot climates are more prone to acquiring novel symbionts that are more tolerant of heat than *Buchnera* itself [29].

In our case, a tendency was apparent that traditionally managed maize crops (small-scale farming and gardens) present in colder regions supporting corn leaf aphids showed overall lower bacterial diversity and were mostly dominated by the primary symbiont *Buchnera* and secondary symbionts *Serratia* and *Wolbachis* (Figure 1B, Table 2).

In relation to host plant effects and secondary symbionts, a study of the pea aphid *Acyrthosiphon pisum* Harris revealed that artificial infection with the symbiont *Hamiltonella defensa* decreased the fitness of the aphids on *Lathyrus* plants (peavines or vetchlings) but not on *Vicia faba* (broad bean), a plant acceptable to the pea aphid genotypes tested. The same study also revealed that by removing the *Hamiltonella* from natural aphid–bacterial associations, an average of 20% decrease in fecundity resulted in aphids on all host plants tested, suggesting ‘universal’ rather than ‘plant-species-specific’ effects of the symbiont [25]. This finding argues for a novel and unexpected role of external environmental factors in governing the endosymbiotic bacterial community of aphids, in turn influencing the aphids themselves in a three-tier cascade, i.e., level of soil nutrition (plus or minus application of synthetic fertilizer and/or pesticides)—diversity of symbiont community within aphid host—aphid host.

In our study in conventionally treated maize crops (large and medium-scale crops), where the dominant treatments were synthetic fertilizers (K, P, and N), the high bacterial diversity (not only endosymbionts but also other taxa as funguses) detected within the infesting corn leaf aphid population was seemingly correlated with synthetic fertilizer applications. Nitrogen especially might have positive effects on aphid performance, probably due to deposition-induced increases in host plant chemistry; the effect of potassium and phosphorus on aphid biology needs further investigation.

The same effect of secondary symbionts on thermal adaptation and nutrition has also been reported [33]. *Serratia* is involved in defense against heat and potentially in aphid nutrition. In our study, *Serratia* was dominant in large-scale crops growth under higher temperatures (north-western region); however, it was also detected in aphids collected from fields with much lower average temperatures (small-scale crops and gardens). The first study describing the two oligophagous aphids, the sugarcane aphid, *Melanaphis sacchari* (Zehntner) and the podocarpus aphid, *Neophyllaphis podocarpi* Takahashi endosymbiotic bacterial diversity related with different geographic regions revealed significant correlation between geographic distances and symbiont communities. Altitude was negatively correlated with the symbiont richness, an interesting finding considering that some secondary symbionts like *Serratia*, *Regiella* and the obligated symbiont *Buchnera* usually protect pea aphids from heat stress by regulating the aphid metabolome [34].

The presence of *Wolbachia* in aphids (i.e, cedar bark aphid, *Cinara cedri* Mimeur) was first detected by Gómez-Valero et al. (2004) who reported that the presence of this microorganism could increase the prevalence of asexual lineages. In our case, *Wolbachia* was present in large-scale field corn leaf samples, but also in samples from small-scale crops, meaning that its presence in natural populations of this aphid species may not be influenced by management or climate, but rather, and more probably, may be the product of an infection by horizontal transfer from some other insect (i.e., parasitoids) [5].

The presence of *Hamiltonella* has been proven to serve as defense against wasp parasitism by arresting development of wasp larvae in pea aphids [6]. In the present study, the presence of this endosymbiont was only detected at a single site (one large-scale crop).

*Rickettsia* was only detected in one aphid colony from a medium-scale crop; the presence of this bacteria could reduce aphid fecundity and longevity (effect similar with *Serratia*) as it has apparently in blue alfalfa aphid, *Acyrthosiphon kondoi* Shinji [35]. For the other symbionts detected, protection against fungal pathogens by *Rickettsia* and *Spiroplasma* has been demonstrated in *A. pisum* [36]. In our study, *Spiroplasma* was detected at low frequency under large-scale management and gardens (Figure 1B), suggesting that its presence is not associated with management systems. Lastly, it is clear that diversity indices also revealed that several bacterial species were found in low frequency, and only a few species from the total bacterial community were dominant

In conclusion, we have demonstrated for the first time the endosymbiotic bacterial diversity of corn leaf aphids under different maize management systems, and most interestingly of all, that variations in such diversity are not wholly associated with management system. Indeed, only the primary symbiont *Buchnera* and secondary symbionts *Serratia* and *Wolbachia* may have a direct effect on aphid adaptations related to management systems. In this light, further research is needed, more especially to test for the possible effect/s of these symbionts in relation to host plant factors, i.e., antifeedants. As a final point, only by the acquisition of such fundamental data can novel pest control approaches be formulated, in this case potential manipulation of the endosymbiotic floral community in natural field populations of pest aphids. By this means, and in the longer term, the reliance on, and current undesirable impact of, synthetic insecticides within the agro-ecosystem will hopefully be reduced.

## Figures and Tables

**Figure 1 microorganisms-10-00939-f001:**
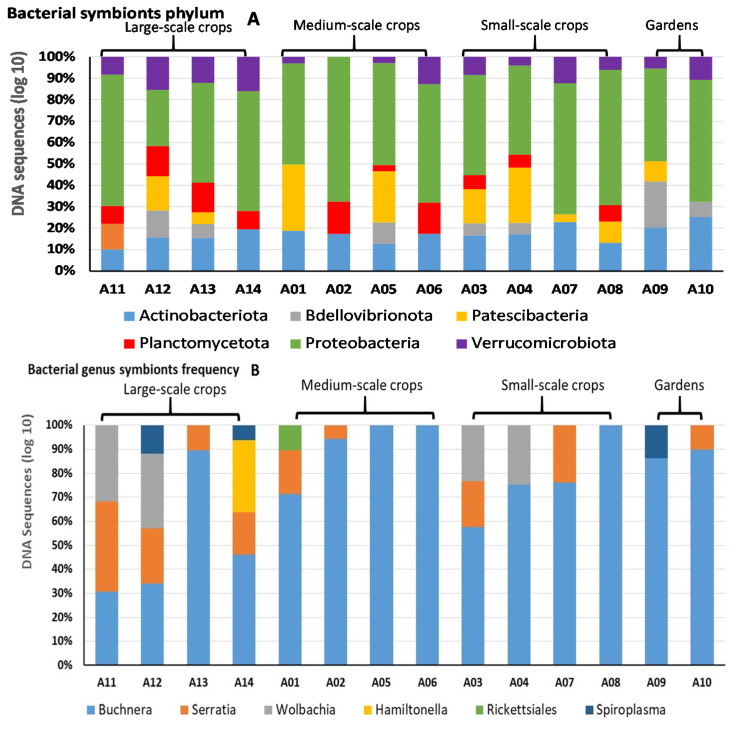
Dominant bacterial phylum variations by sites according to the DNA sequences frequency (**A**) and endosymbiontic bacterial genera variations by sites according to the DNA sequences frequency (**B**).

**Figure 2 microorganisms-10-00939-f002:**
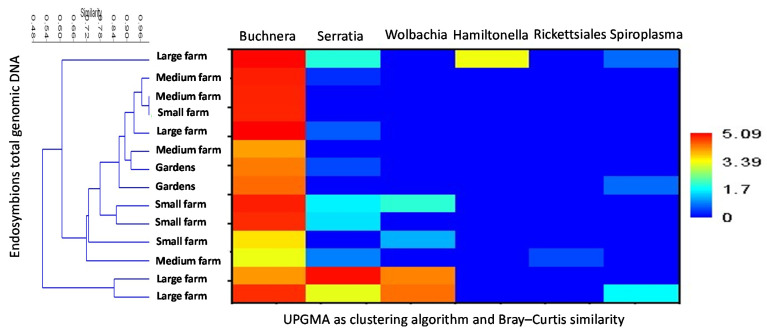
Relative abundance patterns of the endosymbiotic taxa present on all maize fields using UPGMA as clustering algorithm and Bray–Curtis similarity calculation. Endosymbiontic taxa were sorted by their abundance (total genomic DNA) to the differentiation between managements systems. Data are shown for taxa with >0.1% contribution. Blue colours represent no frequency, green a low frequency, yellow medium frequency and red high frequency.

**Figure 3 microorganisms-10-00939-f003:**
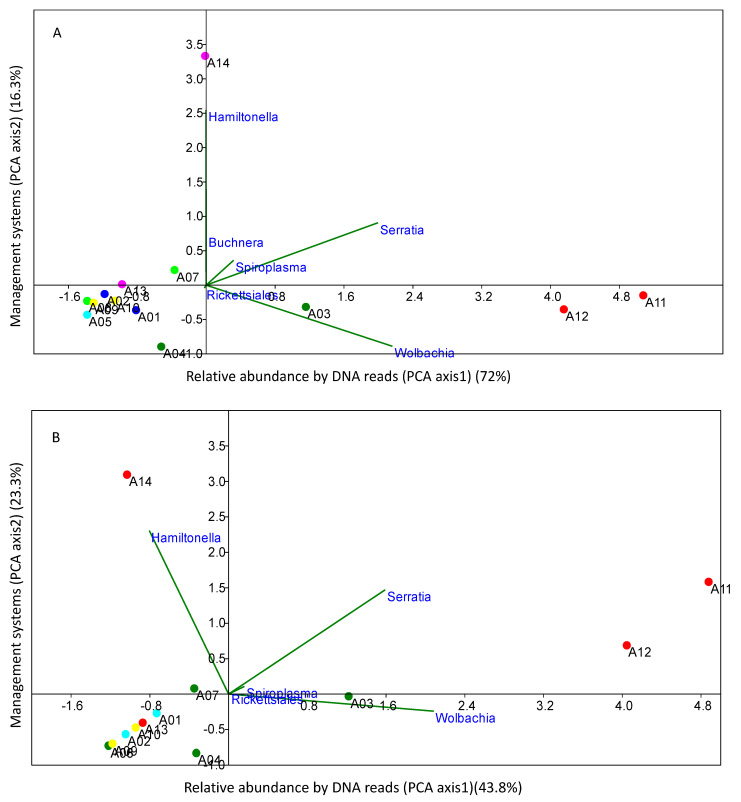
Principal components analyses (PCAs) using the proportion of variation in each PCA axis (bacterial DNA diversity as axis 1 and management as axis 2) explained by the most frequent obligate and facultative bacterial distribution (**A**) and only facultative bacterial distributions (**B**). The average count of each bacterial DNA reads numbers detected and log10 transformed from each sample grouping as response variables as component 1 (PCA axis1) and management systems as component 2 (PCA axis 2) scores were used.

**Table 1 microorganisms-10-00939-t001:** Cropping systems and management used in maize fields assessed.

Cropping System	Management	Sample Code	Symbionts Detected
**Large farms** 2 year old culture, previous year crop sunflower Hybrid: P 9911	**Seed treatment:** Imidacloprid 600 g/L (Seedoprid 600 FS 10 L/*t*) **Herbicide pre-emergent:** Isoxaflutole 225 g/L + Thiencarbazone-methyl 90 g/L + Cyprosulfamide (safener) 150 g/L (Adengo 0.35 L/ha) **Herbicide post-emergent:** Replaced by hoeing. **Fertilizers:** NPK 16:16:16 250 Kg/ha at soil preparation and N 33.3% 250 Kg/ha at hoeing and Foliar fertilizer N (195 g/L), MgSO_4_ (26 g/L), SO_3_ (55 g/L) + microelements (Boron, Copper, Iron, Manganese, Molybdenum, Zinc (14.3 g/L) (Plonvit Active Maize 3 L/ha) **Insecticide:** lambda—cyhalothrin (50 g/L) (Karate Zeon 250 mL/ha) against *Diabrotica* at tasseling	A11 A12	*Buchnera, Serratia, Wolbachia,* *Buchnera, Serratia, Wolbachia, Spiroplasma*
**Large farms** 4 year old culture Hybrid: PR 37 N 01	**Seed treatment:** Fludioxinil + Metalaxyl-M + Thiabendazole+Azoxystrobin **Herbicide****pre-emergent:** Isoxaflutole 225 g/L + Thiencarbazone-methyl 90 g/L + Cyprosulfamide (safener) 150 g/L (Adengo 0.35 L/ha) **Herbicide post-emergent:** Nicosulfuron 40 g/L (Nicogan 40 OD 1 L/ha) and Florasulam 6.25 g/L + Acid 2,4-D EHE 300 g/L (Mustang 0.5 L/ha) **Fertilizers:** NPK 16:16:16 300 Kg/ha at soil preparation	A13 A14	*Buchnera, Serratia* *Buchnera, Serratia, Hamiltonella, Spiroplasma*
**Medium farms** 3 year old culture Hybrid: LG 33.50	**Seed treatment:** Prothioconazole + Metalaxyl. **Herbicide pre-emergent:** Isoxaflutole + Thiencarbazone-methyl. **Fertilizers:** 50 t/ha organic fertilizer at seeding. At 12 leaf stage: Inorganic fertilizer NH_4_NO_3_+CaMg(CO_3_) 2200 kg/ha.	A01 A02	*Buchnera, Serratia, Rickettsiales other than Spiroplasma* *Buchnera, Serratia*
**Medium farms** 5 year old culture Hybrid: LG SHANNON	**Seed treatment:** Prothiocanazole + Metalaxyl. **Herbicide pre-emergent:** Isoxaflutole + Thiencarbazone-methyl. At 4–6 leaf stage Nicosulfuron. **Fertilizers:**70 t/ha organic fertilizer at seeding. At 12 leaf stage: Inorganic fertilizer NH_4_NO_3_+CaMg(CO_3_) 2200 kg/ha.	A05 A06	*Buchnera* *Buchnera*
**Small farms** 1 year old culture, previous year crop wheat and before maize Hybrid: Not known	Only organic fertilizers 50 t/ha before seeding.	A03 A04	*Buchnera, Serratia, Wolbachia* *Buchnera, Wolbachia*
**Small farms** 1 year old culture, previous year crop wheat and before maize Hybrid: Not known	Only organic fertilizers 50 t/ha before seeding.	A07 A08	*Buchnera, Serratia* *Buchnera*
**Gardens** 1 year old culture, previous year crop potato. Hybrid: Not known	No treatment (Control crop)	A09 A10	*Buchnera, Spiroplasma* *Buchnera, Serratia*

**Table 2 microorganisms-10-00939-t002:** Total bacterial diversity indices including endosymbiotic species.

Farming	Large Scale-Farming				Medium-Scale Farming				Small-Scale Farming				Gardens	
Codes	A11	A12	A13	A14	A01	A02	A05	A06	A03	A04	A07	A08	A09	A10
No. of genera	28	166	47	55	47	31	46	59	60	59	63	33	56	50
Simpson	0.920	0.993	0.964	0.969	0.966	0.937	0.966	0.967	0.971	0.971	0.973	0.944	0.969	0.966
Shannon_H	2.922	4.998	3.627	3.733	3.582	3.079	3.578	3.779	3.802	3.754	3.872	3.168	3.716	3.592
Evenness_e^H/S	0.664	0.893	0.800	0.760	0.765	0.701	0.778	0.742	0.746	0.724	0.763	0.720	0.734	0.726

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
