# Peer review of "Endosymbiotic Bacterial Diversity of Corn Leaf Aphid, Rhopalosiphum maidis Fitch (Hemiptera: Aphididae) Associated with Maize Management Systems"

_microorganisms, 2022, doi:10.3390/microorganisms10050939_

Round 1

Reviewer 1 Report

The MS presents very interesting data on aphid endosymbionts, which seems to ba area of increasing interest in many taxa of organisms. From the aphidological point of view I have some minor, following remarks to the current version of the MS:

1) In Introduction (lines 55-60) it might be worthy add info on potential corelation between edosymbionts and ant attendance in aphids (e.g. Depa et al. 2020)

2) Zea mays may be a host plant for several closely related aphid species, including Rhopalosiphum spp., and such differences may influence results on endosymbionts composition e.g. Cinara - how Authors determined species identity? Also, were the sampled aphids somehow purified before DNA extraction? - that would decrease the possibility of DNA contamination, e.g. Jousselin et al. 2016.

3) What does 'contribution' mean at line 179? 

4) There is quite significant difference between study plots A11 and A12 in values of bacteria species and diversity indices. Are you able to provide any explanation of such differences? Could there be any differences in crop management between these sites? 

5) In Supplementary material - what the numbers represent? 

6) line 305, 306 - change font size

Author Response

We would like to thank reviewer work, and we appreciate their nice comments. All the changes were made and color marked in new version of the manuscript. We do hope that the paper can be accepted in its present forms.

Review 1.

The MS presents very interesting data on aphid endosymbionts, which seems to be area of increasing interest in many taxa of organisms. From the aphidological point of view I have some minor, following remarks to the current version of the MS:

Authors: Thank you for your appreciations.

1.In Introduction (lines 55-60) it might be worthy add info on potential correlation between endosymbionts and ant attendance in aphids (e.g. Depa et al. 2020)

Authors: Thank you for this observation. However we did not checked relations between ants and symbionts at this stage. We however planed a more extensive analyses, in which this aspect will be tested. Thus, we would like not to add untested data nor even in Introduction.

2) Zea mays may be a host plant for several closely related aphid species, including Rhopalosiphum spp., and such differences may influence results on endosymbionts composition e.g. Cinara - how Authors determined species identity?

Authors: Thank you, good point. Yes, aphids were identified, and only R. maidis colonies were sampled.

Also, were the sampled aphids somehow purified before DNA extraction? - that would decrease the possibility of DNA contamination, e.g. Jousselin et al. 2016.

Authors: Thank you, Yes, samples were purified before DNA extraction.

3) What does 'contribution' mean at line 179? 

Authors: This means that from the total bacterial community (Table Supplementary materials), approximately 0.1% are endosymbionts.

4) There is quite significant difference between study plots A11 and A12 in values of bacteria species and diversity indices. Are you able to provide any explanation of such differences? Could there be any differences in crop management between these sites? 

Authors: Good statement, thank you. We don’t know yet, as these crops were intensively managed, and we have no influence on this management, further researches are necessary to understand this differences. We did not mentioned this, as we are still running research at that area, and will have much more samples analyzed at the end of this year.

5) In Supplementary material - what the numbers represent? 

Authors: Number means DNA sequences numbers, we specified this in title.

6) line 305, 306 - change font size

Authors: Done, thank you

Reviewer 2 Report

Comments to the Authors

Manuscript: Endosymbiotic bacterial diversity of corn leaf aphid, Rhopalosiphum maidis Fitch (Hemiptera: Aphididae) associated by maize management systems

General comments:

In this study, Csorba et al. accessed total and endosymbiotic bacterial diversity on the corn leaf aphid, Rhopalosiphum maidis, under different management systems. The investigation of the effects of management systems to aphid associated bacterial community is interesting. However, in general, the current status of data presentation and interpretation in the manuscript is rather preliminary. In addition, the English needs to be improved for clarity. Therefore, the manuscript needs significant improvements before publication.  

Major comments:

There are multiple factors involved in the different management systems compared in this study, currently the management systems are considered as a single factor to compare the differences in aphid associated bacterial community. This is OK, however, the analyses of the data are rather preliminary. The authors have speculated that some of the factors (e.g. fertilizer input and/or temperature) may play roles in determining the differences. In the current data collected, the author may perform further comparisons controlling each individual factor. Either the author could perform a multifactor statistical analysis to control the effect of each single factor; or the author could compare sites that are different only in one factor e.g. with/without fertilizers.  

The presentation of results in tables and figures needs improvement for clarity. In table 1, there is a lot of information in the second column, which is hard to compare between different sites, and the sepearation between sample codes in each section is not clear. The last column is unclear, maybe change the title to “symbionts detected”? The hybrids (e.g. P9911 and others) need to be explained for clarity. In table 2, each role of the first column needs explanation for clarity. In Figure 1A, more distinct colors can be used for Planctomycetota and Verrucomicorbiota.

Unclear information in writing:

Line 137: what is “semi fields”?

Line 142: why first instar aphid nymphs were selected to be sampled?

Line 206: “Serratia was present … with lower frequency” . There are many sites that have no Serratia detected, for example A05/06/08/09. The statement needs to be accurate.

Line 219-220: It states “closer similarity between groups from the same management systems are obvious (Figure 2).” Which is not clear to me from what shown in Figure 2, please further explain.

Line 251: “Facultative endosymbiotic Wolbachia dominated large-scale corps …” The statement is not true as shown in Figure 1B, there are no Wolbachia detected in the A13 and A14, which are large-scale crops.

Similar statements need to be carefully checked through the manuscript and rephrased.

English typos, errors and confusing sentences need to be checked throughout the manuscript. For example:

Line 79: “en”

Line 91: “so doing”, “doing so”?

Line 226: “(“ at the beginning of the sentence.

Author Response

We would like to thank reviewer work, and we appreciate their nice comments. All the changes were made and color marked in new version of the manuscript. We do hope that the paper can be accepted in its present forms.

Review 2

General comments:

In this study, Csorba et al. accessed total and endosymbiotic bacterial diversity on the corn leaf aphid, Rhopalosiphum maidis, under different management systems. The investigation of the effects of management systems to aphid associated bacterial community is interesting. However, in general, the current status of data presentation and interpretation in the manuscript is rather preliminary. In addition, the English needs to be improved for clarity. Therefore, the manuscript needs significant improvements before publication.  

Authors: Thank you for this observations, we tried to make requested changes.

 Major comments:

There are multiple factors involved in the different management systems compared in this study, currently the management systems are considered as a single factor to compare the differences in aphid associated bacterial community. This is OK, however, the analyses of the data are rather preliminary. The authors have speculated that some of the factors (e.g. fertilizer input and/or temperature) may play roles in determining the differences. In the current data collected, the author may perform further comparisons controlling each individual factor. Either the author could perform a multifactor statistical analysis to control the effect of each single factor; or the author could compare sites that are different only in one factor e.g. with/without fertilizers.  

Authors: Principal Components Analyses (PCA) were used to identify the proportion of variation in each PCA axis (bacterial DNA diversity and management) that was explained by the most frequent obligate and facultative bacterial distribution. The aver-age count of each bacterial DNA reads numbers detected and log10 transformed from each sample grouping as response variables as component 1 (PCA axis1) and management systems as component 2 (PCA axis 2) scores were used.

The presentation of results in tables and figures needs improvement for clarity. In table 1, there is a lot of information in the second column, which is hard to compare between different sites, and the separation between sample codes in each section is not clear. The last column is unclear, maybe change the title to “symbionts detected”? The hybrids (e.g. P9911 and others) need to be explained for clarity.

Authors: Thank you, we made changes in table 1, trying to make it clearer. The column 2 contains important information, we tied to make it clearer now. About hybrids, these names are given by the company, we do not have another information about maize hybrids, however if clarified what needed here, we can do more explanation.

In table 2, each role of the first column needs explanation for clarity.

Authors: These are general diversity indices used in such analyses, we mentioned in table caption. Please clarify what another explanations would be necessary here?

In Figure 1A, more distinct colors can be used for Planctomycetota and Verrucomicorbiota.

Authors: Good observation, thank you, we changed colors.

Line 137: what is “semi fields”?

Authors: Semi-field here means that inside each field two smaller site were sampled. Now we changed this in text.

Line 142: why first instar aphid nymphs were selected to be sampled?

Authors: This is a general method at aphids, because of their asexual reproduction, older nymphs and adults, already having embryos, so for bacterial analyses more than one individual and its symbionts are tested if not first instars (with no embryos in their body).

Line 206: “Serratia was present … with lower frequency” . There are many sites that have no Serratia detected, for example A05/06/08/09. The statement needs to be accurate.

Authors: Thank you, agree, we changed this sentence.

Line 219-220: It states “closer similarity between groups from the same management systems are obvious (Figure 2).” Which is not clear to me from what shown in Figure 2, please further explain.

Authors: Thank you, we made changes as follows: Using UPGMA as clustering algorithm and Bray-Curtis similarity, it can be detected some effects of the management, groups from the same management systems shows similarity, but also some differences can be detected (Figure 2).

Line 251: “Facultative endosymbiotic Wolbachia dominated large-scale corps …” The statement is not true as shown in Figure 1B, there are no Wolbachia detected in the A13 and A14, which are large-scale crops.

Authors: Thank you, agree, we changed this sentence

English typos, errors and confusing sentences need to be checked throughout the manuscript. For example:

Authors: Done, the whole manuscript has been checked by Prof. Loxdale, one of the coauthor in this paper

Line 79: “en”

Authors: Done, deleted

Line 91: “so doing”, “doing so”?

Authors: Done, changed

Line 226: “(“ at the beginning of the sentence.

Authors: Done, deleted

Reviewer 3 Report

The idea that management of the maize on which aphids live could affect their obligate endosymbionts is intriguing to me, so the paper caught my attention. It is well written.

Is it possible management did not affect endosymbionts, but just increased the abundance of non-endosymbionts such that the relative proportion of endosymbionts appeared to change? Would it be possible to see direct measurement of the levels of Buchnera? Use qPCR or something similar to see if Buchnera levels are the same in aphids from different crops. 

A problem with the paper is the small scale and garden crops are also from a colder climate. You mention in the introduction that temperature can affect aphid endosymbionts, so how do you know if the results in your paper are due to climate or management, or even just random population differences?

I would have liked more explanation of the PCA analysis. How did you figure out the relative effects of the different species? What does it all mean?

Comments:
Last sentence of the abstract is too vague. I would prefer you discuss it here, and maybe delete some other text from the abstract if you need to make room.

Line 63 "fin" should be "in"
64 delete "in other words"
71 missing "Glover" for Aphis gossypii
93 "Other" to "Another"
108 "marks the presentation"
109 missing period at end of sentence
115 need a "?" after "diversity"
200-202 This sentence has a few different grammar/spelling errors that make it hard to understand. Can you say it more simply? Example: "Aphids collected from large-scale crops site A12 all had four species of bacterial endosymbionts, while aphids from the other locations often had fewer. Only the obligate symbiont Buchnera was found in all samples."
206 "another" to "other"    [the word "another" means "an other," and refers to an individual "other," so if referring to multiple others you say "other." Why English says "another" instead of "an other," I do not know.]

Figure 1 is great, but looking at it I cannot tell if these differences are significant.
Figure 2 The A## values on the tree are hard to read, especially the yellow ones. Color-code them into the 4 crop types (large, medium, small, garden), and use dark colors.

226 The first "(" can be deleted, and I think a ")" is missing after "axis 2"
229 "symbiont distributions" or "symbionts' distributions"
259-260 the sentence seems to end abruptly. "here abiotic constraints" what? A predicate clause with a verb is missing.
265 ", but some"
269-270 you use "these" to refer to hosts and symbionts in the same sentence. I would replace the first one with "the hosts" and the second with "the old symbionts"
278 delete comma after "regions"
305-306 the font size is bigger for some reason
325 "our study"
333 "research is"

Author Response

We would like to thank reviewer work, and we appreciate their nice comments. All the changes were made and color marked in new version of the manuscript. We do hope that the paper can be accepted in its present forms.

Review 3

The idea that management of the maize on which aphids live could affect their obligate endosymbionts is intriguing to me, so the paper caught my attention. It is well written.

Authors: Thank you, we appreciate the reviewer effort in improving our manuscript.

Is it possible management did not affect endosymbionts, but just increased the abundance of non-endosymbionts such that the relative proportion of endosymbionts appeared to change? Would it be possible to see direct measurement of the levels of Buchnera? Use qPCR or something similar to see if Buchnera levels are the same in aphids from different crops. 

Authors: Thank you, we agree, we do planed such measurement, however the Illumina is so expensive, we only have found for these analyses. We also need to add, that several project applications are under review, and in case of positive decision a much bigger data collection and additional analyses will be made.

A problem with the paper is the small scale and garden crops are also from a colder climate. You mention in the introduction that temperature can affect aphid endosymbionts, so how do you know if the results in your paper are due to climate or management, or even just random population differences?

Authors: Thank you, good point. We don’t know at this moment the right answer, and we do not want to speculate. This year we will have found to run a much more detailed sampling and analyses, and will be able to have several other samples from colder, but also from cooler climate including all management systems. As the Illumina is very expensive, we were limited at this point.

I would have liked more explanation of the PCA analysis. How did you figure out the relative effects of the different species? What does it all mean?

Authors: Done: Principal Components Analyses (PCA) were used to identify the proportion of variation in each PCA axis (bacterial DNA diversity and management) that was explained by the most frequent obligate and facultative bacterial distribution. The aver-age count of each bacterial DNA reads numbers detected and log10 transformed from each sample grouping as response variables as component 1 (PCA axis1) and management systems as component 2 (PCA axis 2) explained by most dominant bacteria related to each management (these dominant bacteria DNA reads numbers, and also separate only Buchera dominance) scores were used.

Comments:
Last sentence of the abstract is too vague. I would prefer you discuss it here, and maybe delete some other text from the abstract if you need to make room.

Authors: Agree thank you, we deleted this sentence as we also considered fictional at this stage of the research.

Line 63 "fin" should be "in"

Authors: Done

64 delete "in other words"

Authors: Done

71 missing "Glover" for Aphis gossypii

Authors: Done

93 "Other" to "Another"

Authors: Done

108 "marks the presentation"

Authors: Done

109 missing period at end of sentence

Authors: Done

115 need a "?" after "diversity"

Authors: Done

200-202 This sentence has a few different grammar/spelling errors that make it hard to understand. Can you say it more simply? Example: "Aphids collected from large-scale crops site A12 all had four species of bacterial endosymbionts, while aphids from the other locations often had fewer. Only the obligate symbiont Buchnera was found in all samples."

Authors: Agree, thank you. Done

206 "another" to "other"    [the word "another" means "an other," and refers to an individual "other," so if referring to multiple others you say "other." Why English says "another" instead of "an other," I do not know.]

Authors: Agree, thank you. Done

Figure 1 is great, but looking at it I cannot tell if these differences are significant.

Authors: Thank you for this observation. These figures do not represents significant differences, only frequencies are compared, no significances presented.

Figure 2 The A## values on the tree are hard to read, especially the yellow ones. Color-code them into the 4 crop types (large, medium, small, garden), and use dark colors.

Authors: Thank you. We made changes in figure 2.

229 "symbiont distributions" or "symbionts' distributions"

Authors: Done

259-260 the sentence seems to end abruptly. "here abiotic constraints" what? A predicate clause with a verb is missing.

Authors: Done

265 ", but some"

Authors: Done

269-270 you use "these" to refer to hosts and symbionts in the same sentence. I would replace the first one with "the hosts" and the second with "the old symbionts"

Authors: Done

278 delete comma after "regions"

Authors: Done

305-306 the font size is bigger for some reason

Authors: Done

325 "our study"

Authors: Done

333 "research is"

Authors: Done